# Acquired Hemophilia A after SARS-CoV-2 Immunization: A Narrative Review of a Rare Side Effect

**DOI:** 10.3390/vaccines12070709

**Published:** 2024-06-25

**Authors:** Roberto Castelli, Antonio Gidaro, Roberto Manetti, Paolo Castiglia, Alessandro Palmerio Delitala, Pier Mannuccio Mannucci, Samantha Pasca

**Affiliations:** 1Department of Medical Surgical and Pharmacology, University of Sassari, 07100 Sassari, Italy; rmanetti@uniss.it (R.M.); paolo.castiglia@uniss.it (P.C.); aledelitala@uniss.it (A.P.D.); 2Department of Biomedical and Clinical Sciences Luigi Sacco, Luigi Sacco Hospital, University of Milan, 20157 Milan, Italy; 3Fondazione Istituto di Ricerca e Cura a Carattere Scientifico Ca’ Granda Ospedale Maggiore Policlinico, Angelo Bianchi Bonomi, Hemophilia and Thrombosis, 20122 Milan, Italy; piermannuccio.mannucci@policlinico.mi.it; 4U.O. Immunohematology and Transfusion—APSS of Trento, 38122 Trento, Italy; sampasca27@gmail.com

**Keywords:** acquired hemophilia A, SARS-CoV-2, COVID-19, vaccine, mRNA vaccines, BNT162b2, mRNA-1273, viral vector vaccine, Vaxzevria, inactivated vaccine, CoronaVac

## Abstract

Acquired hemophilia A (AHA) is a rare bleeding disorder (1.4 per million inhabitants per year) caused by neutralizing antibodies against factor VIII. Although uncommon, these autoantibodies can cause a high rate of morbidity and mortality. Several conditions are linked with AHA; based on an EACH2 study, 3.8% of AHA could be connected to infection. In the last four years, most humans have contracted the SARS-CoV-2 infection or have been vaccinated against it. Whether or not COVID-19 immunization might induce AHA remains controversial. This review aims to evaluate the evidence about this possible association. Overall, 18 manuscripts (2 case series and 16 case reports) were included. The anti-SARS-CoV-2 vaccination, as also happens with other vaccines, may stimulate an autoimmune response. However, older individuals with various comorbidities are both at risk of developing AHA and of COVID-19-related morbidity and mortality. Therefore, the COVID-19 vaccine must always be administered because the benefits still outweigh the risks. Yet, we should consider the rare possibility that the activation of an immunological response through vaccination may result in AHA. Detailed registries and prospective studies would be necessary to analyze this post-vaccine acquired bleeding disorder, looking for possible markers and underlying risk factors for developing the disease in association with vaccination.

## 1. Introduction

Acquired hemophilia A (AHA) is an uncommon bleeding disorder that arises from neutralizing antibodies against coagulation factor VIII. Its incidence stands at approximately 1.4 cases per million individuals annually [1]. Although rare, these autoantibodies can lead to a high rate of morbidity and mortality. Severe bleeds occur in up to 90% of cases, and the mortality rate ranges from 8% to 22%. The bleeding pattern of AHA is different from that of congenital hemophilia A. Most patients with factor VIII autoantibodies develop hemorrhages into the skin, muscles, or soft tissues, in mucous membranes (such as epistaxis, gastrointestinal and urologic bleeds), retroperitoneal hematomas, and postpartum bleeding. In contrast, hemarthrosis, a typical feature of congenital factor VIII deficiency, is uncommon [2,3,4,5]. Hemorrhages can have serious or life-threatening consequences, as the disease may present with excessive bleeding after trauma or surgery, or as cerebral hemorrhage. AHA becomes more common with age, and it is rare for children to experience this condition. It has been estimated that children under 16 have a lower incidence rate of approximately 0.045 per million per year than older adults over 85 years, with an incidence rate of 14.7 per million per year. Whether the age-related increase is due to immunosenescence-inducing autoreactive immune responses or the rise of comorbidities potentially inducing AHA remains unexplained. 

The prevalence of this autoimmune disorder may be significantly underestimated, especially among elderly patients. In approximately 50% of cases, factor VIII autoantibodies are present in patients with no relevant concurrent diseases. The remaining cases may be linked to autoimmune diseases, underlying hematologic or solid cancers, infections, medication usage, and dermatologic conditions such as pemphigus and psoriasis. Acquired hemophilia A also occurs in about 5% of pregnancies, typically after childbirth. Several conditions are linked with AHA (Table 1) [6,7,8,9,10]. 

Due to higher disease awareness and diagnosis, the incidence of AHA has surged to 6 cases per million. Infections can be associated with AHA; based on an EACH2 study, 3.8% of AHA were connected to infection [11].

With this introduction on AHA, a new strain of coronavirus was identified as the cause of a series of pneumonia cases in Wuhan, a city in the Hubei Province of China in late 2019. The virus rapidly spread, leading to an epidemic throughout China, followed by many other countries. In February 2020, the World Health Organization (WHO) officially named COVID-19, the coronavirus disease 2019, responsible for severe acute respiratory syndrome. From the beginning of 2021, vaccines against SARS-CoV-2 became available worldwide. Thanks to their prevention, the world has controlled this pandemic. Exposure to the vaccine could theoretically lead to autoimmune conditions such as AHA. Because whether or not SARS-CoV-2 immunization might induce AHA remains controversial, this review aims to evaluate the evidence about this possible association. 

## 2. Pathophysiology of Aha

Acquired clotting factor inhibitors are autoantibodies that hinder the activity or accelerate the clearance of coagulation factors. Function-blocking antibodies are a common type of antibodies that often affect factor VIII. Factor VIII serves as a cofactor in the intrinsic tenase complex. It plays a crucial role by providing a scaffold for bringing together factor IXa and factor X. This interaction activates factor X to Xa on the surface of platelets and endothelial cells. This process occurs via binding to phospholipids. When circulating in the bloodstream, factor VIII is stabilized by binding to von Willebrand factor (VWF), and acquired factor VIII antibodies can interfere with both of these functions. Moreover, certain types of factor VIII antibodies possess proteolytic activity that breaks down proteins and thus increases their removal from the bloodstream [5].

Factor VIII is an essential precursor protein with a substantial molecular weight of 330 kDa. Through proteolytic processing, its domain structure changes into heavy and light chain heterodimers. Acquired factor VIII inhibitors typically target the A2, A3, or C2 domains. The A2 and A3 domains play crucial roles in binding factor VIII to factors IXa and X. On the other hand, the C2 domain facilitates the binding of factor VIII to phospholipids on activated platelets, endothelial cells, and von Willebrand factor. Autoantibodies that arise in patients with AHA are predominantly of the IgG class, mainly IgG1 and IgG4, and are non-complement fixing and non-precipitating [12].

## 3. SARS-CoV-2 Infection, Autoimmunity and Autoimmune Diseases

The development of autoimmunity has been associated with various viral infections, other than SARS-CoV-2, such as Epstein-Barr virus (EBV), cytomegalovirus (CMV), parvovirus B19, and hepatitis B (HBV) and C viruses (HCV) [13]. Multiple epitopes (hexapeptides) have been discovered in both the SARS-CoV-2 spike (S) protein and human organ proteins. This discovery highlights the potential for autoimmunity development through cross-reactivity [13]. This may occur due to molecular mimicry and/or bystander activation. The proteins involved in this process include ribosomal proteins, methyltransferases, cytokines such as interleukin 7 (IL-7), lysosomal, sodium channel, cell adhesion, and myosin proteins [13]. During a viral infection, molecular cross-reactivity may occur, resulting in the production of autoantibodies and the activation of autoreactive T-cells. This can also lead to the emergence of hidden antigens and the spread of epitopes. Similar cross-reactivity-related autoimmune phenomena have been observed during previous coronavirus epidemics, such as SARS-CoV-1 and MERS-CoV [14]. Some peptides present in specific proteins of SARS-CoV-2 may have the ability to cross-react with peptides in lung alveolar surfactant proteins, and molecular mimicry might potentially contribute to developing pediatric inflammatory multisystem syndrome [15]. Moreover, the hexapeptide cross-reactivities identified could also be involved in the clinical symptoms associated with COVID-19 [15]. Histone lysine methyltransferases may contribute to developing neurodevelopment disorders, convulsions, and behavioral disorders [16]. On the other hand, IL-7 plays a vital role in regulating the immune system, and its deficiency can lead to severe lymphopenia [17]. COVID-19 is commonly associated with both neuropsychiatric abnormalities and lymphopenia [18]. In pediatric inflammatory multisystem syndrome, at least six protein epitopes of the SARS-CoV-2 virus cross-react with the inositol triphosphate 3 kinase C (ITPKC) Kawasaki antigen, strongly indicating the involvement of molecular mimicry in the development of this disease [13]. Research on COVID-19 has shown that this viral infection triggers acute autoimmune and autoinflammatory mechanisms, as indicated by inflammation during the disease, the inflammatory multisystem syndrome, and clinical and radiological phenomena. The lung injury caused by SARS-CoV-2, known as ARDS, shares many similarities with the acute exacerbation of interstitial lung disease (ILD) associated with autoimmune diseases. Autoantibodies were detected in 20–50% of patients with pneumonia associated with COVID-19 [19]. More than 20 autoantibodies have been identified with SARS-CoV-2 infection. The most common ones are antinuclear antibodies (ANA) and antibodies against coagulation cascade components such as antiphospholipid (APLA), anti-prothrombin, and anti-heparin. However, SARS-CoV-2 infection has also produced anti-citrullinated protein (ACPA), anti-neutrophil cytoplasmic antigen (ANCA), anti-SS-A/Ro antibodies, and rheumatoid factor [13]. Several studies have examined the changes in tissues caused by SARS-CoV-2. Samples of tissues obtained during autopsy from 18 COVID-19 patients who died a revealed significant infiltration of T-cells, including various T-cell subclasses in the lungs. The kidney, liver, intestinal wall, and pericardium also showed small cellular infiltrates. The prevailing cell type in all cases was the CD8+ T lymphocyte [20]. Autoimmune syndromes involving the central nervous system, such as the Guillain-Barré and Miller-Fisher syndrome, as well as immune thrombocytopenia (ITP), systemic lupus erythematosus (SLE), and Kawasaki disease (KD), have been described [13]. 

## 4. SARS-CoV-2 Vaccination Reactions, Autoimmunity and Autoimmune Diseases

Different types of vaccines have been developed to combat the SARS-CoV-2 pandemic and its consequences. All activate the immune system against the spike protein, directly encoded (mRNA-based vaccines) or introduced (viral vector or inactivated virus vaccines) in the host organism. 

The mRNA and recombinant protein vaccines are associated with cases of myocarditis and pericarditis more commonly than anticipated in young male individuals who received the Pfizer COVID-19 vaccine and the Moderna COVID-19 vaccine [21]. Cases were also observed during the phase 3 trials of the Novavax COVID-19 vaccine [21]. Surveillance data also suggest the possibility of an increased risk following the Janssen/Johnson & Johnson COVID-19 vaccine [21]. Because these cases are infrequent and typically mild, the benefits of vaccination far outweigh the small increased risk [21].

The vaccines that use adenovirus vectors, such as Ad26.COV2. S (Janssen/Johnson & Johnson COVID-19 vaccine) and ChAdOx1 nCoV-19/AZD1222 (AstraZeneca COVID-19 vaccine) have been associated with cases of Guillain-Barré syndrome (GBS) [22]. However, no similar association has been observed with the mRNA COVID-19 vaccines. Cases of GBS, including recurrent cases, have also been reported in individuals infected with SARS-CoV-2. Observational data suggests that the risk of developing GBS after a SARS-CoV-2 infection is higher than that associated with vaccination [22].

Both the ChadOx1 nCoV-19/AZD1222 (AstraZeneca COVID-19 vaccine) and Ad26.COVID-19 (Janssen/Johnson & Johnson COVID-19) have been associated with the risk of uncommon types of thrombotic events with thrombocytopenia. No similar risk has been identified with mRNA vaccines, making them a safe choice. Many of these cases are associated with autoantibodies directed against the platelet factor 4 (PF4) antigen, similar to those found in patients with autoimmune heparin-induced thrombocytopenia (HIT) [23]. The syndrome is known as Vaccine-associated Immune Thrombotic Thrombocytopenia (VITT), while some refer to it as thrombosis with thrombocytopenia syndrome (TTS). It has been observed that individuals who develop adenovirus infection without receiving any vaccine also experience similar findings. VITT is triggered by antibodies that target platelet factor 4 (PF4) or CXCL4. These antibodies are IgGs that activate platelets through low-affinity platelet FcγIIa receptors present on the platelet surface, which bind the Fc portion of IgG [24]. Platelet and neutrophil activation activate the coagulation system, leading to clinically significant thrombotic complications. The mechanism by which the implicated vaccines trigger the development of new antibodies (and/or immune stimulation of preexisting antibodies) remains uncertain. A recent model suggests a two-hit process in which the vaccine stimulates neoantigen formation (first hit) along with a systemic inflammatory response (second hit), which together lead to the production of anti-PF4 antibodies [24]. Vaccine components such as virus proteins, proteins from the HEK3 cell line from which the vaccine is produced, and free DNA may bind to PF4 and alter its conformation, thus generating a neoantigen [23]. VITT can manifest in common sites of venous thromboembolism, like pulmonary embolism or deep vein thrombosis (DVT) in the leg. However, a defining aspect of this syndrome is the occurrence of blood clots in atypical locations, including the splanchnic (splenic, portal, mesenteric) veins, adrenal veins (posing a risk for adrenal failure), and cerebral and ophthalmic veins [23]. Individuals with a history of heparin-induced thrombocytopenia (HIT) or thrombosis should avoid adenoviral vaccines and consider receiving a different type of COVID-19 vaccine. Arterial thrombosis, such as ischemic stroke (often in the middle cerebral artery) and peripheral arterial occlusion, has also been observed in individuals with concurrent venous thrombosis [23].

## 5. Aim

This review aims to analyze the impact of SARS-CoV-2 vaccination on the development of AHA, the people most frequently affected, and the potential causes of AHA as a complication of vaccination. 

## 6. Methods

The following review was conducted following the Preferred Reporting Items for Systematic Reviews and Meta-Analyses (PRISMA) model [25]. The literature search encompassed clinical studies, case reports, reviews, abstracts, and all scientific articles related to acquired hemophilia A (AHA) occurring after anti-SARS-CoV-2 vaccination and published on PubMed from January 2021 to April 2024. The search terms “acquired hemophilia A” and “acquired hemophilia A” were used, combined with the Boolean operator “AND”, as well as terms such as “anti-SARS-CoV-2 vaccination” OR “anti-SARS-CoV-2 vaccine”. These terms were searched explicitly within the articles’ titles and/or abstracts. The review included publications that met the following criteria: (1) manuscripts concerning patients with AHA after anti-SARS-CoV-2 vaccination; (2) manuscripts written in English. No report analyzed in this review included patients with current or previous SARS-CoV-2 infection. The PRISMA Flowchart of search is shown in Figure 1.

## 7. Results

Seventy-six publications met our search requirements; after evaluation, 36 were excluded because they were published before 2021, i.e., before the beginning of mass vaccination against SARS-CoV-2. Eleven manuscripts were subsequently removed. They did not deal with this investigated topic, nine because they were reviews, correspondences, or replies, and two because they were not written in English. Overall, 18 manuscripts (2 case series and 16 case reports) were included in this review. 

The cases of AHA were almost all identified after the administration of mRNA vaccines—only one being reported after an inactivated virus vaccine. No patient had or previously had SARS-CoV-2 infection. The first few reports of AHA following SARS-CoV-2 vaccination date back to 2021 [26,27,28]. Overall, five patients experienced this autoimmune coagulation disorder; two different cases [26,28] were reported after the second dose of BNT162b2 (Pfizer-BioNTech), and the case series by Cittone et al. [27] described three cases that occurred after the second dose of mRNA-1273 (Moderna) (two patients), or after the first dose of this vaccine (one patient). As reported in other studies on AHA, the autoimmune bleeding disorder affected in these reports equally between males and females aged 67–86 years. A few cases presented concomitant cardiovascular diseases; one had diabetes and a previous cancer. The outcomes were generally favorable with a spontaneous clearance of the autoantibodies; only one patient died following gallbladder rupture and related severe arterial bleeding. 

The most numerous cases were reported in 2022, when vaccinations peaked worldwide. Sixteen patients [29,30,31,32,33,34,35,36,37,38,39,40,41] experienced this complication, only one after receiving a viral vector vaccine [41]. In all of the remaining cases, AHA occurred after BNT162b2 (Pfizer-BioNTech), equally divided between the first, second, and third doses (booster). Two patients aged 39 and 45 suddenly showed signs and symptoms of this hemorrhagic disease after the first and booster doses, respectively, both being previously in good health with no history of bleeding or chronic diseases [32,39]. Thus, AHA is likely attributable to the mRNA vaccine’s effect, which triggered a response against the spike protein and autoimmunity against coagulation factor VIII.

Although most cases witnessed the resolution of the spontaneous hemorrhagic disease without further complications, two patients subsequently developed severe events, one a sarcoma and one a bullous pemphigoid [35,36]. An additional patient experienced rapidly resolved gastrointestinal bleeding [38], and two patients died due to sepsis unrelated to the vaccine [30,41]. Excluding the two youngest reported patients, the others were similar to the cases reported in the previous year and the AHA studies at large [11,42] and had various comorbidities. The last two cases included in this review are from 2023; one refers to a 67-year-old man [43] who experienced AHA after the first dose of mRNA-1273 (Moderna), and the second illustrates the case of a 69-year-old man [44] who developed a left inner thigh ecchymosis after the second dose of the inactivated virus vaccine CoronaVac (SinoVac). As in the case of Plüß et al. [36] and also in that described by Franchini & Focosi [43], the patient subsequently developed a sarcoma that led to death. This patient, in addition to a previous diagnosis of pulmonary sarcoidosis and rheumatoid arthritis, had experienced two different episodes of AHA, the first, idiopathic, in 2011 and the subsequent relapse in 2020 after contracting the SARS-CoV-2 infection, both managed with steroids and cyclophosphamide. A second case of previous SARS-CoV-2 infection was described by Happaerts & Vanassche [41]. In this case, the patient had been hospitalized for having contracted COVID pneumonia 14 months before receiving the vaccine and was immediately treated with amoxicillin/acid clavulanate and Anakinra 100 mg per day. Instead, Emna et al. [44] reported no consequences in patients without concomitant disease.

Management of AHA involves treating acute bleeding and eradicating the inhibitor. Almost a third of patients (8/23) aged between 39 and 95 years described in this review [26,30,31,32,38,39,44] did not require hemostatic treatment with bypassing agents, recombinant porcine factor VIII or emicizumab. A 95-year-old patient [31] was administered a dose of recombinant factor VIII 2000 IU as a precaution, while a second one [38] was treated for a few days with plasma-derived factor VIII and von Willebrand factor enriched (pdFVIII/vWF) for gastrointestinal bleeding, which subsequently resolved. Eleven patients [27,30,33,34,36,37,40,43] needed a brief treatment with rFVIIa, while only one was treated with aPCC [28]; a combination of rFVIIa and aPCC [29,35] was used in two patients and a combination of rFVIIa and emicizumab in the only patient with AHA after viral vector vaccination [41]. Immunosuppressive therapy with corticosteroids was started in 22/23 patients [26,27,28,29,30,31,32,33,34,35,36,37,38,39,40,43,44], and rituximab was used as the first-line attempt to eradicate the inhibitor only in the case reported by Happaerts & Vanassche [41]. In comparison, rescue therapy was described in seven patients [27,28,30,31,32,36,38]. Concomitant treatment with cyclophosphamide was carried out in six patients [29,30,35,36,37,40]. 

The complete list of reports included in this review is shown in Table 2.

## 8. Discussion

Acquired hemophilia A (AHA) is an uncommon bleeding disorder that impacts both genders equally. It is estimated to affect between 1 and 6 individuals per million annually, although it may be underestimated because not all cases are recognized and recorded. AHA is due to the loss of tolerance toward the body’s factor VIII and the development of autoantibodies against factor VIII. This entire process can be attributed to the anergy or loss of antigen-specific T and B lymphocytes and the elimination of self-reactive T lymphocytes during the maturation of the immune system [45]. 

The occurrence of AHA demonstrates a dual-phase trend, characterized by an initial peak in young women during pregnancy or postpartum, followed by a more pronounced surge in the older population. This bleeding disorder is idiopathic in half of the cases, while the remaining cases are associated with autoimmune, dermatological, or infectious diseases, postpartum or pregnancy, cancer, and drugs [46]. 

Our review aimed to underline, similarly to what was reported by Franchini & Focosi [43], how the cases of AHA associated with anti-SARS-CoV-2 vaccines increased over time as a function of the more significant number of people vaccinated. The subjects most at risk were older people, often presenting with various co-morbidities, i.e., the so-called “fragile subjects” for whom vaccination is recommended. However, before subjecting them to repeated immunizations, a careful evaluation of the risks and benefits, the underlying pathologies, and the bleeding history of each patient should be performed. Almost all of the reported cases involved subjects vaccinated with mRNA-based vaccines. The adverse event occurred after administering the first, second, or booster doses, even though a slight preponderance of AHAs developing after the second dose was noticed. In a recent publication, Yasmin et al. [47] reported severe adverse events after administering this vaccine. Although the majority of these events were thrombotic or cardio-vascular, cerebral hemorrhage unrelated to AHA occurred in 8% of reported cases in BNT162b2 (Pfizer-BioNTech) vaccinated subjects and 3% of mRNA-1273 (Moderna) vaccinated subjects. Therefore, hemorrhagic events as a consequence of vaccinations cannot be ruled out. 

National and international guidelines recommend prompt treatment of AHA-induced bleeding by using bypassing agents (activated prothrombin complex concentrate (aPCC) or activated recombinant factor VII (rFVIIa)) or recombinant porcine factor VIII (rpFVIII). Recombinant or plasma-derived factor VIII can instead be used in case of a shortage of these products or when the inhibitor titer is low (<5 BU/mL) [4,48]. Recently, the bispecific monoclonal anti-body emicizumab, initially licensed to treat congenital hemophilia A, was also authorized to treat AHA in Japan [49]. 

The risk of thrombotic events following the use of bypassing agents, especially in elderly subjects with multiple co-morbidities, must always be considered before starting a hemostatic therapy [50]. In this review, we observed how, in approximately one-third of the cases, the resolution of the acute events occurred without using any anti-hemorrhagic therapy [26,30,31,32,38,39,44]. AHA was a transient episode triggered by exposure to the vaccine, which did not cause consequences in any affected patients. No thrombotic event was observed in those who, for short periods, required treatment with bypassing agents or emicizumab, used in a single case as a prophylactic treatment after the resolution of acute bleeding with rFVIIa [41]. Promptly ensuring normal hemostasis is crucial. To eradicate inhibitors, the use of corticosteroids alone or in combination with cyclophosphamide is recommended. In cases where immunosuppressive therapy fails, rituximab, a potent monoclonal antibody targeting B cells, is highly recommended [4,48]. All of the subjects described in this review were treated with corticosteroids [26,27,28,29,30,31,32,33,34,35,36,37,38,39,40,43,44], some in combination with cyclophosphamide [29,30,35,36,37,40]. Rituximab was instead used in about a fourth of patients [27,28,30,31,32,36,38,41]: a high percentage if we consider what has been published in the various registers, where this medicine is not used [42] or rarely used only as a rescue treatment [51].

Only two patients had had a previous SARS-CoV-2 infection; the first patient [41] had contracted pneumonia 14 months before vaccination. Treatment was immediate, and the event was resolved. It is not easy to hypothesize that an event so distant in time could have influenced the onset of the AHA in any way. In the second case [48], the vaccine seems to have triggered a relapse of AHA exactly, as had previously happened following the SARS-CoV-2 infection. A similar case of relapse immediately after COVID infection was reported by Marumo et al. [52].

The basis of AHA is the activation of the immune system, which produces autoantibodies against endogenous factor VIII, similar to what happens in autoimmune diseases. In the present review, a few cases had rheumatological diseases; the anti-SARS-CoV-2 vaccination, therefore, seems to have triggered a further activation of the immune system with the development of new autoantibodies, this time against coagulation factor VIII. This mechanism of action is the same as that reported in AHA cases due to immune checkpoint inhibitor therapy [53]. 

The causality between the administration of vaccines against SARS-CoV-2 and the onset of this bleeding disorder cannot be proven with certainty, even if the cases of Hosoi et al. [39] and Soliman et al. [32] occurred in two young patients without any typically underlying comorbidity. Detailed registries and prospective studies will be necessary to analyze this post-vaccine acquired bleeding disorder, looking for possible markers and underlying risk factors for developing the disease.

## 9. Limitations

This review’s primary limitation is that it could analyze only case reports and case series. The COVID-19 vaccines were developed in record time, and vaccination was rapidly implemented everywhere to protect especially the most fragile subjects from severe disease. No ad hoc registers or prospective studies were set up to analyze the onset of an AHA among the possible side effects. Given the results of this review and the published reports, it is hoped that registers or studies to evaluate this aspect will also be established in the future.

## 10. Conclusions

The anti-SARS-CoV-2 vaccination, as also happens with other vaccines, stimulates an immune response. Older individuals with co-morbidities are most at risk of developing AHA, owing to the activation of the immune system against endogenous factor VIII. Therefore, attention must be paid when subjecting these patients to multiple vaccination cycles because there is a risk of activating an immunological response.

## Figures and Tables

**Figure 1 vaccines-12-00709-f001:**
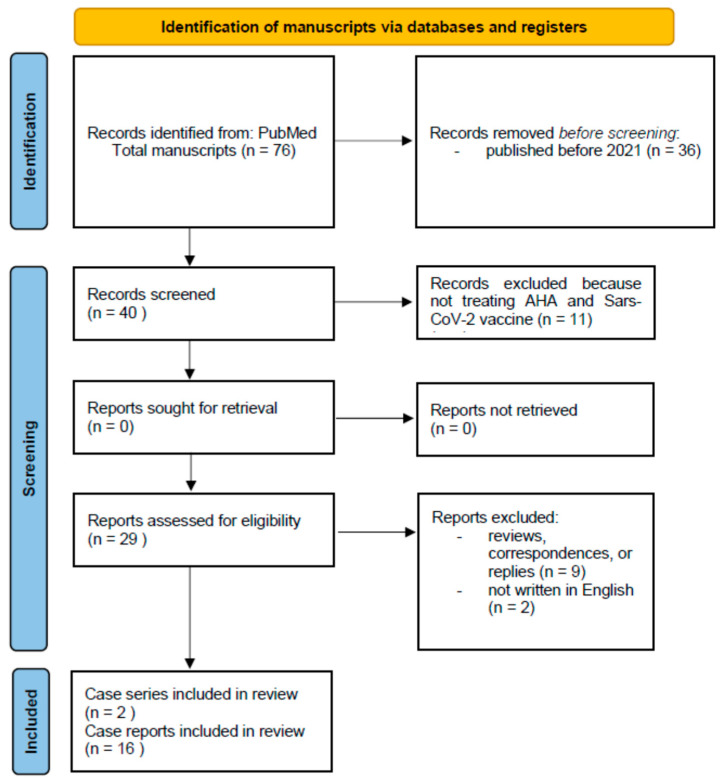
PRISMA Flowchart of search.

**Table 1 vaccines-12-00709-t001:** Diseases and clinical conditions associated with AHA (adapted from AICE Recommendations 2020).

Diseases or Clinical Conditions	Characteristics
**Oncologic diseases**	Multiple myeloma, lymphomas, monoclonal gammopathy of uncertain significance (MGUS), myelofibrosis, myelodysplasia
**Rheumatic diseases**	Rheumatoid arthritis, systemic lupus erythematosus, Sjogren’s syndrome, Goodpasture’s syndrome, temporal arteritis, myasthenia gravis, thyroiditis, multiple sclerosis
**Infectious diseases**	SARS-CoV-2, Epstein Barr virus, hepatitis B/C viruses, Human Immunodeficiency virus
**Dermatological diseases**	Psoriasis, pemphigus
**Pregnancy or Puerperium**	Within 1-4 months of delivery or miscarriage
**Drugs**	Some beta-lactam antibiotics, chloramphenicol, sulfonamides, clopidogrel, nonsteroidal anti-inflammatory drugs (NSAIDs), fludarabine, interferon alpha
**Other diseases**	Asthma, chronic obstructive pulmonary disease, acute hepatitis

**Table 2 vaccines-12-00709-t002:** A complete list of AHA reports from January 2021 to January 2024 in order of publication. DM2: diabetes mellitus type 2; HCV: hepatitis C virus; BPH: benign prostate hypertrophy; RBC: red blood cells; TXA: tranexamic acid; CS: Corticosteroids; rFVIIa: recombinant FVII activated; Cyc: Cyclophosphamide; aPCC: activated prothrombin complex concentrate; rFVIII: recombinant FVIII; DDAVP: desmopressin; FVIII/vWF: plasma-derived FVIII with von Willebrand factor; NA: not available.

Ref.	Author(s), Year	Patient(s) (Sex; Age)	Vaccine Type	Dose(s) n	Days after Vaccination	Clinical Manifestations	Concomitant Diseases	Acute Treatments	Outcomes
[26]	Radwi & Farsi, 2021	1 (M; 69yrs)	BNT162b2 (Pfizer-BioNTech)	2	9	Spontaneous bruises on arms and legs	Prostate adenocarcinoma, DM2, hypertension	CS	Resolved without sequelae
[27]	Cittone et al., 2021	1 (M; 85 yrs) 1 (F; 86yrs) 1 (F; 72yrs)	mRNA-1273 (Moderna) mRNA-1273 (Moderna) mRNA-1273 (Moderna	2 2 1	Immediately after 2nd dose 21 10	Multiple hematomas on the right thigh, joint bleeding (knees) Traumatic hemothorax Table Extensive spontaneous cutaneous bruising	Peripheral artery disease, coronary bypass Aortic valve stenosis Arterial disease	rFVIIa, aPCC, CS, RTX rFVIIa, aPCC, CS rFVIIa, TXA, CS, RTX	Active arterial bleeding after gall rupture, death Resolved without sequelae NA
[28]	Farley et al., 2021	1 (M; 67yrs)	BNT162b2 (Pfizer-BioNTech)	2	21	Large hematoma posterior left leg	Asymptomatic pulmonary sarcoidosis, hypertension	aPCC, rFVIIa, CS, RTX	Resolved without sequelae
[29]	Lemoine et al., 2022	1 (M; 70yrs)	mRNA-1273 (Moderna)	1	8	Extensive right upper limb bruising	Rheumatic polymyalgia, previous HCV infection	aPCC, rFVIIa, CS, Cyc	Resolved without sequelae
[30]	Leone et al., 2022	1 (M; 86yrs) 1 (F; 73yrs) 1 (M; 67yrs) 1 (M; 77yrs)	BNT162b2 (Pfizer-BioNTech) BNT162b2 (Pfizer-BioNTech) BNT162b2 (Pfizer-BioNTech) BNT162b2 (Pfizer-BioNTech)	2 2 2 2	14 26 49 52	Disseminated hematomas, severe anemia Spontaneous tongue, jaw, and right knee hematomas Tongue hematoma Hematuria, severe anemia	Rheumatic polymyalgia Rheumatoid arthritis, Sjogren syndrome None Bladder cancer relapse	RBC, CS CS rFVIIa, CS, Cyc rFVIIa, CS, RTX	Resolved without sequelae Resolved without sequelae Resolved without sequelae Sepsis, respiratory complication, death
[31]	Murali et al., 2022	1 (F; 95yrs)	BNT162b2 (Pfizer-BioNTech)	2	21	Spontaneous bruising on limbs	Dementia, hypertension, depression, congestive cardiac failure, previous breast cancer	RBC, rFVIII, CS, RTX	Resolved without sequelae
[32]	Soliman et al., 2022	1 (F; 39yrs)	BNT162b2 (Pfizer-BioNTech)	1	10	Hematuria	None	CS, RTX	Resolved without sequelae
[33]	Vuen et al., 2022	1 (M; 80yrs)	BNT162b2 (Pfizer-BioNTech)	1	14	Ecchymosis at limbs, severe anemia	DM2, hypertension, dyslipidemia, chronic kidney disease, glaucoma in both eyes, ischemic stroke	rFVIIa, TXA, CS	Resolved without sequelae
[34]	Al Hennawi et al., 2022	1 (M; 75yrs)	BNT162b2 (Pfizer-BioNTech)	2	90	Soft tissue ecchymoses, compartment syndrome, anemia	Hypertension, dyslipidemia, coronary artery disease	rFVIIA, DDAVP, CS, RTX	Resolved without sequelae
[35]	Fu et al., 2022	1 (M; 77yrs)	mRNA-1273 (Moderna)	2	21	Bilateral legs, feet, and ankles ecchymosis	NA	aPCC, rFVIIa, CS, Cyc	Bullous pemphigoid
[36]	Plüß et al., 2022	1 (M; 72yrs)	BNT162b2 (Pfizer-BioNTech)	Booster	9	Arms, left leg, and trunk bruises	BPH, carpal tunnel syndrome	rFVIIa, CS, Cyc, RTX	Pleomorphic dermal sarcoma
[37]	Rashid et al., 2022	1 (M; 75yrs)	BNT162b2 (Pfizer-BioNTech)	Booster	6	Both thighs and back bruises	DM2	RBC, rFVIIa, CS, Cyc	Resolved without sequelae
[38]	Melmed et al., 2022	1 (F; 61yrs)	mRNA-1273 (Moderna)	2	14	Inner thigh bruising, anemia with dyspnea	Rheumatoid arthritis	RBC, FVIII/vWF	Gastrointestinal bleeding
[39]	Hosoi et al., 2022	1 (F; 45yrs)	mRNA-1273 (Moderna) *1st and 2nd doses Pfizer BioNTech*	Booster	14	Subcutaneous hemorrhage	None	CS	Resolved without sequelae
[40]	Duminuco et al., 2023	1 (M; 71yrs)	BNT162b2 (Pfizer-BioNTech)	2	8	Hemothorax and hemoperitoneum in the abdomen, extending from the pelvic area to the right thigh	None	RBC, rFVIIa, CS, Cyc	Resolved without sequelae
[41]	Happaerts & Vanassche, 2022	1 (M; 75yrs)	Vaxzevria ChAdOx1 (AstraZeneca)	1	10	Multiple hematomas, hemorrhagic bullous pemphigoid, gastrointestinal ulcer	Chronic kidney disease, hypertension, DM2, polyneuropathy, chronic foot ulcer	rFVIIa, Emicizumab, RTX, CS	Atrial fibrillation, acute kidney injury, and methicillin-sensitive *Staphylococcus aureus* sepsis, death
[43]	Franchini & Focosi, 2023	1 (M; 67yrs)	mRNA-1273 (Moderna)	1	22	Anemia	Rheumatoid arthritis, pulmonary sarcoidosis	rFVIIa, RTX	Liposarcoma, death
[44]	Emna et al., 2023	1 (M; 69yrs)	CoronaVac (SinoVac)	2	30	Left inner thigh ecchymosis	None	RBC, TXA, CS	Resolved without sequelae

## Data Availability

Not applicable.

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
