# Peer review of "Acquired Hemophilia A after SARS-CoV-2 Immunization: A Narrative Review of a Rare Side Effect"

_vaccines, 2024, doi:10.3390/vaccines12070709_

Round 1
Reviewer 1 Report
Comments and Suggestions for Authors
This review entitled “Acquired Hemophilia A after SARS-CoV-2 immunization: actual 2 risk or not?” by Castelli and Gidaro et. al. focuses on an important phenomenon after SARS-Cov-2 immunization. They were able to find 18 papers in the literature and conclude that the vaccination may stimulate an autoimmune response and, given the age range and the burden of comorbid conditions among those experiencing this adverse effect, suggest that great caution should be implemented in vaccinating these individuals and the booster doses. Although this is an interesting topic, I have the following concerns:
Major comments:
1. as the authors have mentioned, this is an extremely rare adverse effect. The total incidence of acquired hemophilia A is 1.4 per million. It should be clearly mentioned in the paper and the abstract as well that a causal association has not been established, and the benefits of vaccination still outweigh the risks like this. You don’t want to convey the wrong message to the public.
2. Line 192: The PRISMA checklist should accompany the paper. The current version in some instances (e.g the title) does not comply with certain sections from the checklist (PRISMA 2020 checklist — PRISMA statement (prisma-statement.org))
3. Table 2: It is important to highlight the clinical management in the table as well.
4. It is important to assess the causality based on the literature in the discussion section of the manuscript. Do you think that these indivuals would not have otherwise developed acquired Hemophilia A had they not been vaccinated?
Minor comments:
1. Table 1. Page 2: Infections should also be included in the table with examples from the literature from other pathogens.
2. Line 61: The term a new strain is accurate.
3. Lines 95-98: needs reference
4. Lines 105-108: needs reference
Comments on the Quality of English LanguageNeeds revision to improve readability
Author Response
Dear Editor,
Enclosed is our revised manuscript entitled “Acquired Hemophilia A after Sars-CoV-2 immunization: actual risk or not?.” We want to thank the Editor and the reviewers for their constructive criticism, which substantially improved our paper. We hope that the paper will be suitable for publication. We have made a point-by-point answer to the referees’ comments below.
Moreover, a complete English revision was performed.
Answers to Reviewer 1
Reviewer 1 comment: This review entitled “Acquired Hemophilia A after SARS-CoV-2 immunization: actual 2 risk or not?” by Castelli and Gidaro et. al. focuses on an important phenomenon after SARS-Cov-2 immunization. They were able to find 18 papers in the literature and conclude that the vaccination may stimulate an autoimmune response and, given the age range and the burden of comorbid conditions among those experiencing this adverse effect, suggest that great caution should be implemented in vaccinating these individuals and the booster doses.
Response to Reviewer 1 comment: We thank the reviewer for his/her positive comment.
Reviewer 1 comment: 1. as the authors have mentioned, this is an extremely rare adverse effect. The total incidence of acquired hemophilia A is 1.4 per million. It should be clearly mentioned in the paper and the abstract as well that a causal association has not been established, and the benefits of vaccination still outweigh the risks like this. You don’t want to convey the wrong message to the public.
Response to Reviewer 1 comment: We thank the reviewer for his/her correction. In the revised version, we changed the abstract to underline that vaccination's benefits still outweigh AHA's risks.
Reviewer 1 comment: 2. Line 192: The PRISMA checklist should accompany the paper. The current version in some instances (e.g the title) does not comply with certain sections from the checklist (PRISMA 2020 checklist — PRISMA statement (prisma-statement.org))
Response to Reviewer 1 comment: We thank the reviewer for his/her correction. This is not a systematic review but a narrative one. For this reason, the title does not mention a systematic review. In the revised version, we have added the PRISMA checklist, in which we have detailed the search criteria
Reviewer 1 comment: 3. Table 2: It is important to highlight the clinical management in the table as well.
Response to Reviewer 1 comment: We thank the reviewer for his/her correction. In the revised version, we have added in Table 2 the different AHA treatments. The duration of the various treatments and dosages were not included in the table because they were clearly expressed only in a few reports.
Reviewer 1 comment: 4. It is important to assess the causality based on the literature in the discussion section of the manuscript. Do you think that these indivuals would not have otherwise developed acquired Hemophilia A had they not been vaccinated?
Response to Reviewer 1 comment: We thank the reviewer for his/her correction. In the revised version, we explained that the causality between vaccine and AHA is not definable.
Reviewer 1 comment: 1. Table 1. Page 2: Infections should also be included in the table with examples from the literature from other pathogens.
Response to Reviewer 1 comment: We thank the reviewer. In the revised version, we have added in Table 1 the infectious diseases involved in AHA development.
Reviewer 1 comment: 2. Line 61: The term a new strain is accurate.
Response to Reviewer 1 comment: We thank the reviewer.
Reviewer 1 comment: 3. Lines 95-98: needs reference
Response to Reviewer 1 comment: We thank the reviewer. In the revised version, we added a reference.
Reviewer 1 comment: 4. Lines 105-108: needs reference
Response to Reviewer 1 comment: We thank the reviewer. In the revised version, we added a reference.
Reviewer 2 Report
Comments and Suggestions for Authors
The title of the article is sensationalist and purports to answer a question that is ultimately not answered as a result of the review. I suggest modifying the title, making reference to the fact that hemophilia could be a side effect, rather than talking about risk.
The review has a major limitation. The type of publications reviewed are case reports and case series, none of the articles reviewed contemplate an appropriate method to establish causality between vaccination and hemophilia. Moreover, none of the case reports mention the fact of NOT having had COVID-19 before or at the time of vaccination to eliminate this confusion, given that the disease itself could be causal for hemophilia. This fact, however, is mentioned by the authors as a possible cause in section 3.
Thus, it should at least be pointed out in the discussion that in order to link the vaccine studied with hemophilia, larger epidemiological studies with the appropriate method are required to establish association and, subsequently, with sufficient evidence, causality.
Other suggestions for improvement are related to:
Abstract: in line 23-24 the phrase "a hemorrhagic disorder caused by the activation of the immune system against endogenous factor VIII" repeats what was mentioned in line 14. The abstract should try to answer the question and conclude that further research is required to establish causality, especially ensuring that there is no confusion with having had SARS-CoV-2 infection recently.
The method does not mention the quality and validation criteria of the selected studies, nor how many reviewers participated and whether there was review of the articles by one or more reviewers, or whether criteria were established to select articles when there were contradictory or disparate evaluations between two reviewers.
In the discussion there are no limitations described. A major limitation is the type of studies reviewed, which are only case reports. Additionally, among the limitations of the review is publication bias, having reviewed only English-language journals.
The conclusions should answer the research question. Specifically, it is not possible to establish with confidence a relationship between COVID-19 vaccination and hemophilia. Studies with appropriate methodologies are required. However, it is possible to identify a tendency that in a certain profile of hemophilic patients may be a side effect of vaccination.
Author Response
Dear Editor,
Enclosed is our revised manuscript entitled “Acquired Hemophilia A after Sars-CoV-2 immunization: actual risk or not?.” We want to thank the Editor and the reviewers for their constructive criticism, which substantially improved our paper. We hope that the paper will be suitable for publication. We have made a point-by-point answer to the referees’ comments below.
Moreover, a complete English revision was performed.
Answers to Reviewer 2
Reviewer 2 comment: The title of the article is sensationalist and purports to answer a question that is ultimately not answered as a result of the review. I suggest modifying the title, making reference to the fact that hemophilia could be a side effect, rather than talking about risk.
Response to Reviewer 2 comment: We thank the reviewer. According to your suggestion, we changed the title to “Acquired Hemophilia A after Sars-CoV-2 immunization: a side effect”
Reviewer 2 comment: The review has a major limitation. The type of publications reviewed are case reports and case series, none of the articles reviewed contemplate an appropriate method to establish causality between vaccination and hemophilia. Moreover, none of the case reports mention the fact of NOT having had COVID-19 before or at the time of vaccination to eliminate this confusion, given that the disease itself could be causal for hemophilia. This fact, however, is mentioned by the authors as a possible cause in section 3. Thus, it should at least be pointed out in the discussion that in order to link the vaccine studied with hemophilia, larger epidemiological studies with the appropriate method are required to establish association and, subsequently, with sufficient evidence, causality.
Response to Reviewer 2 comment: We thank the reviewer. In the revised version, we clarified that no patient had or had previously had Sars-Cov-2 infection. Given the limited number of cases, having trials or registries that best describe the hemorrhagic event and its management is impossible. To date, no retrospective registers have been created in this regard, but it would be desirable to have some in the future to answer the unresolved questions.
Reviewer 2 comment: Other suggestions for improvement are related to:
Abstract: in line 23-24 the phrase "a hemorrhagic disorder caused by the activation of the immune system against endogenous factor VIII" repeats what was mentioned in line 14. The abstract should try to answer the question and conclude that further research is required to establish causality, especially ensuring that there is no confusion with having had SARS-CoV-2 infection recently.
Response to Reviewer 2 comment: We thank the reviewer. In the revised version, we removed the duplicate sentence. The abstract was changed according to both reviewer's suggestions.
Reviewer 2 comment: The method does not mention the quality and validation criteria of the selected studies, nor how many reviewers participated and whether there was review of the articles by one or more reviewers, or whether criteria were established to select articles when there were contradictory or disparate evaluations between two reviewers.
Response to Reviewer 2 comment: We thank the reviewer. This is not a systematic review but a narrative one. For this reason, we didn’t specify how many reviewers participated. Moreover, we did not consider it necessary to include the validation criteria of the selected studies as they are all case reports/case series published in indexed journals that use peer reviewers but present the typical limitations of case reports.
Reviewer 2 comment: In the discussion there are no limitations described. A major limitation is the type of studies reviewed, which are only case reports. Additionally, among the limitations of the review is publication bias, having reviewed only English-language journals.
Response to Reviewer 2 comment: We thank the reviewer. In the revised version, we have inserted a "limitations" paragraph.
Reviewer 2 comment: The conclusions should answer the research question. Specifically, it is not possible to establish with confidence a relationship between COVID-19 vaccination and hemophilia. Studies with appropriate methodologies are required. However, it is possible to identify a tendency that in a certain profile of hemophilic patients may be a side effect of vaccination.
Response to Reviewer 2 comment: We thank the reviewer. The revised version specifies that more evidence is needed to establish a causal relationship between COVID-19 vaccination and hemophilia.
Round 2
Reviewer 1 Report
Comments and Suggestions for Authors
Figure one legend should not be indicated as checklist. It is the flowchart.
Comments on the Quality of English LanguageMinor edit
Author Response
Dear Editor,
Enclosed is our revised manuscript entitled “Acquired Hemophilia A after Sars-CoV-2 immunization: a rare side effect.” We want to thank the Editor and the reviewers for their constructive criticism, which substantially improved our paper. We hope that the paper will be suitable for publication. We have made a point-by-point answer to the referees’ comments below.
Moreover, a complete English revision was performed.
Answers to Reviewer 1
Reviewer 1 comment: Figure one legend should not be indicated as checklist. It is the flowchart. Response to Reviewer 1 comment: We thank the reviewer for his/her correction. Figure one legend is now correctly indicated as the flowchart.
Reviewer 2 Report
Comments and Suggestions for Authors
The authors improved the manuscript with the suggestions provided. However, the main limitation has not been resolved, which is that among the cases and case series there are patients who previously presented SARS-CoV-2 infection.
I made a random review of some of the original reports of the review, finding, for example, that reference 46 is of a case that did present the disease prior to vaccination. Therefore, the authors' assertion in lines 202 (No report analyzed in this review included patients with current or previous Sars- 202 CoV-2 infection) and 215-216 that “no patients had prior SARs-CoV-2 infection” is not true. It seems to me that it is necessary that the authors revisit the original cases and effectively eliminate, at least, those in which previous infection is described.
Author Response
Dear Editor,
Enclosed is our revised manuscript entitled “Acquired Hemophilia A after Sars-CoV-2 immunization: a rare side effect.” We want to thank the Editor and the reviewers for their constructive criticism, which substantially improved our paper. We hope that the paper will be suitable for publication. We have made a point-by-point answer to the referees’ comments below.
Moreover, a complete English revision was performed.
Answers to Reviewer 2
Reviewer 2 comments: The authors improved the manuscript with the suggestions provided.
Response to Reviewer 2 comment: We thank the reviewer for his/her positive comment.
Reviewer 2 comment: However, the main limitation has not been resolved, which is that among the cases and case series there are patients who previously presented SARS-CoV-2 infection.
I made a random review of some of the original reports of the review, finding, for example, that reference 46 is of a case that did present the disease prior to vaccination. Therefore, the authors' assertion in lines 202 (No report analyzed in this review included patients with current or previous Sars- 202 CoV-2 infection) and 215-216 that “no patients had prior SARs-CoV-2 infection” is not true. It seems to me that it is necessary that the authors revisit the original cases and effectively eliminate, at least, those in which previous infection is described.
Response to Reviewer 2 comment: We thank the reviewer for his/her correction. In the revised version, we reviewed all the individual cases reported in the case reports and case series. Only two patients had previously contracted Sars-CoV-2 infection. We have described their characteristics in the results and the discussion.